# Epigenetic Regulation in Breast Cancer: Insights on Epidrugs

Ayoung Kim [1,†], Kyumin Mo [1,†], Hyeonseok Kwon [1,†], Soohyun Choe [1], Misung Park [1,2], Woori Kwak [1,2,*] and Hyunho Yoon [1,2,*]

1 Department of Medical and Biological Sciences, The Catholic University of Korea, Bucheon 14662, Republic of Korea
2 Department of Biotechnology, The Catholic University of Korea, Bucheon 14662, Republic of Korea
* Correspondence: woori@catholic.ac.kr (W.K.); hyoon@catholic.ac.kr (H.Y.)
† These authors contributed equally to this work.

**Abstract:** Breast cancer remains a common cause of cancer-related death in women. Therefore, further studies are necessary for the comprehension of breast cancer and the revolution of breast cancer treatment. Cancer is a heterogeneous disease that results from epigenetic alterations in normal cells. Aberrant epigenetic regulation is strongly associated with the development of breast cancer. Current therapeutic approaches target epigenetic alterations rather than genetic mutations due to their reversibility. The formation and maintenance of epigenetic changes depend on specific enzymes, including DNA methyltransferases and histone deacetylases, which are promising targets for epigenetic-based therapy. Epidrugs target different epigenetic alterations, including DNA methylation, histone acetylation, and histone methylation, which can restore normal cellular memory in cancerous diseases. Epigenetic-targeted therapy using epidrugs has anti-tumor effects on malignancies, including breast cancer. This review focuses on the importance of epigenetic regulation and the clinical implications of epidrugs in breast cancer.

**Keywords:** epigenetic regulation; epidrug; breast cancer; DNMT; HDAC





## 1. Introduction

Cancer is a complex disease and a major health concern worldwide. Among gynecological cancers, breast and ovarian cancers are representative diseases that are fatal. Breast cancer is the most common cancer among women. The incidence of breast cancer has increased by 0.5% per year over the past decade, whereas mortality has decreased by 1.3% per year owing to advances in diagnostic methods, increased early detection, and advances in drugs and treatments [1]. Breast cancer can be diagnosed by histological methods and can be divided into several types depending on their molecular markers: hormone receptor (HR), including estrogen receptor (ER) and progesterone receptor (PR), and human epidermal growth factor receptor 2 (HER2). HR+/HER2- patients who are HR-positive and HER2-negative represent the most common subtypes of breast cancer, followed by those with HR+/HER2+, HR-/HER2+, and HR-/HER2-. Recently, gene screening data by next-generation sequencing (NGS) has demonstrated hereditary associations involving *BRCA1/2, CDH1, BRIP1,* and *PTEN* between breast cancer and ovarian cancer [2,3]. In particular, *BRCA1/2*, breast cancer-associated genes 1 and 2, are the most well-known tumor suppressors in breast cancer [4]. Mutations in the *BRCA1/2* gene lead to dysfunction of the normal cell cycle and a high risk of breast cancer [5]. Other important genes for breast cancer include *EGFR2/HER2* and *HER1*, which are well-known oncogenes in many cancers. Overexpression of *EGFR* promotes cell proliferation, cell invasion, angiogenesis, and prevention of cancer cell apoptosis via the dysfunction of downstream signaling involving PI3K, Ras-Raf-MAPK, and JNK [4,6]. EGFR and related signaling molecules are currently important targets for the treatment of breast cancer, leading to improved patient survival.

Epigenetic alterations are a major cause of breast cancers. Epigenetics is the study of molecules and mechanisms that can perpetuate alternative gene activity states in the context of the same DNA sequence [7]. Epigenetic mechanisms include the regulation of gene transcription, genomic stability, and maintenance of normal cell growth, development, and differentiation. The most important characteristic of epigenetic alterations is that such changes does not alter the genetic material and reprogram genomic information. Three main mechanisms are known: DNA methylation, histone modification, and non-coding RNAs (ncRNAs). They are critical regulators of cellular immunity, which is mediated via the regulation of gene expression and transcription in specific cells and tissues.

Epigenetic alterations are critical drivers of tumor initiation and progression. During tumorigenesis, a variety of epigenetic changes occur, including genome-wide loss of DNA methylation, local hypermethylation of CpG promoters of tumor suppressor genes, broad changes in histone modification, and deregulation of the ncRNA network. In addition, alterations in epigenetic enzymes significantly contribute to abnormal patterns of DNA methylation and histone modification in cancer. The identification of mutations in the DNA methylation machinery serves as a starting point for the development of more specific diagnostic and prognostic tools for cancer [8]. Aberrant DNA methylation patterns (both hyper- and hypomethylation) have been studied in many different types of cancer, including prostate, breast, gastric, liver, lung, glioblastoma, and leukemia. histone acetyltransferase (HAT) and histone deacetylases (HDAC) are well translocated and mutated in hematological malignancies and solid tumors. DNMT3A is also frequently mutated in acute myeloid leukemia (AML) and T-cell lymphoma [9]. DNMT3A mutations are found in hematopoietic stem cells in the blood of patients with AML [10]. In addition, the deregulation of ncRNAs is significantly associated with cancer progression.

Epigenetic drugs (epidrugs) control the enzymes necessary for maintenance and promote the destruction of transcriptional and post-transcriptional changes. Epidrugs are chemical agents that reactivate epigenetically silenced tumor suppressors and DNA repair genes by altering the DNA and chromatin structure. Epidrugs not only modulate gene expression; they can also affect the regulation and dysregulation of genes [11]. The major strategy for epidrugs involves inhibiting DNMTs and HDACs. A DNMT inhibitor (DNMTi) inserts itself between DNA base pairs to inhibit CpG dinucleotide methylation [12]. DNMTi, a first-generation epidrug, is a pyrimidine analog present in DNA during replication and activates the DNA damage response, resulting in cell death. HDAC inhibitor (HDACi) decreases the activation of $Zn^{2+}$-dependent HDAC enzymes. HDACi suppresses pattern changes mediated by histone acetylation and facilitates restoration to the normal state. As cancer initiation and progression are strongly associated with epigenetic and genetic alterations, epidrugs are promising for drug development targeting cancer epigenetic genes. In this review, we discuss the importance of epigenetic regulation and the potential of epidrugs in breast cancer.

## 2. Regulatory Enzymes of Histone Methylation and Acetylation

DNA methylation was the first epigenetic alteration to be discovered in humans in the early 1980s. DNA methylation by the covalent transfer of a methyl group to the C-5 position of the cytosine ring is catalyzed by DNA methyltransferases (DNMTs). DNA methylation influences genomic imprinting, X chromosome inactivation, chromosome stability, suppression of transposable elements, aging, and numerous diseases, such as cancer and autoimmune diseases. It can be lost or driven by the enzymatic process of dioxygenases, namely ten-eleven translocation (TET) proteins, which promote the oxidation of methylcytosine to hydroxymethylcytosine during the mammalian development [13]. In mammals, DNA methylation occurs predominantly at CpG dinucleotides, and the genomes are generally CpG-depleted and methylated [14]. Histone modifications mostly occur at specific positions involving the amino-terminal and carboxy-terminal histone tails. Histone acetylation occurs on lysine residues by balancing the action of two enzymes, HAT and HDAC. The most well-known histone modification is histone methylation of arginine and

lysine residues and both are correlated with various human diseases such as cancer [15]. The most well-known histone modification involves the methylation of arginine and lysine residues. Unlike histone acetylation, methylation does not alter physical interactions between DNA and histones. Histone methylation is also regulated by 'writer' lysine methyltransferases (KMTs) and 'reader' and 'eraser' lysine demethylases (KDMs) [16]. ncRNAs are functional but do not translate into functional proteins that control gene expression post-transcriptionally and at the transcriptional level via the organization and modification of chromatin.

## 3. Epigenetic Regulation in Breast Cancer

### 3.1. Role of Noncoding RNA in Epigenetic Regulation

Breast cancer is associated with several epigenetic changes. ncRNAs, particularly miRNAs, are involved in the post-transcriptional regulation of breast cancer tumorigenesis, progression, and metastasis. miRNA, a ncRNA with a length of 17–25 nt, can regulate gene expression in normal and abnormal cells, including cancer cells. Polymerase II promotes transcription of primary miRNAs. The Drosha and Dicer complex processes primary miRNAs into pre-miRNAs, allowing mature forms of miRNAs to regulate gene expression at the post-transcriptional level. In cancer, miRNAs aberrantly regulate genes owing to CpG hypermethylation in miRNA genes or dysregulation of miRNA biosynthesis processes [17]. Abnormal miRNA processes are involved in every stage of cancer, from tumorigenesis to metastasis in breast cancer. For example, miRNA profile data have shown that tumor suppressor miRNAs, such as miR-4458 associated with the SOC1 signaling pathway, are suppressed, whereas oncogenic miRNAs such as miR-214, related to the PI3K/Akt/mTOR pathway, are upregulated in breast cancer [18]. Dysregulation of these miRNAs leads to the manifestation of hallmarks of cancer, including cell proliferation, metastasis, apoptotic response, hypoxia, and angiogenesis [19].

Both miRNAs and lncRNAs are hallmarks of breast cancer. LncRNAs, which are ncRNAs with lengths ranging from 200 nt to 100 kb, can silence target genes by promoting DNA methylation or histone modification. Dysregulation of these processes can lead to tumorigenesis [20]. For example, the lncRNA GAS5, which acts as a tumor suppressor via the regulation of various tumor suppressor proteins, such as PTEN, PDCD4, OKK2, FOXO1, and SUFU, is markedly downregulated in breast cancer. In addition, GAS5 expression is suppressed by promoter methylation in triple-negative breast cancer (TNBC) [21], suggesting that lncRNAs play key roles in aggressiveness of breast cancer.

### 3.2. Estrogen-Related Epigenetic Mechanisms

Breast cancer subtypes can be identified based on epigenetic mechanisms. In breast cancer, ER-mediated epigenetic changes are regulated by transcription factors and co-regulators. Estrogen regulates the mitotic and epigenetic mechanisms of mammary gland formation. Estrogen can be classified into five types: estrone, estrogen, 17-β estradiol, estriol, and estrone-sulfate [22]. Estradiol (E2) acts as an initiator of breast cancer. Treatment with E2 induces tumorigenesis of breast cancer in vitro via anchor-independent growth, loss of ductulogenesis in collagen, and invasiveness in Matrigel [23]. Many co-regulators, such as the p160 family, protein arginine methyltransferases, p300, and some mediator complexes, are recruited to chromatin for E2 stimulation, which is enhanced by ERα [24,25].

The Wnt signaling and ERα pathways are linked through polycomb proteins [26]. The polycomb group protein enhancer of zeste homolog 2 (EZH2) is an important factor that can directly interact with ERα and β-catenin to link estrogen and Wnt signaling [26], suggesting that Wnt signaling is highly associated with tumorigenesis and metastasis in ER-positive breast cancer. Wnt antagonist genes, such as *SERP* and *DKK*, are regulated by epigenetic silencing via DNA methylation in breast cancer. β-catenin, which activates Wnt-mediated genes, is continuously activated by methylation of Wnt antagonistic genes, thereby increasing the rate of regeneration and proliferation of stem cells, resulting in poor prognosis and disease recurrence [24,27]. For example, *DKK3* promoter methylation

was found in 78% of patients with primary breast cancer, and these patients had a poorer prognosis and higher metastasis rates than patients without *DKK3* methylation [28]. In addition, epigenetic silencing of *DKK3* resulted in lymph node metastasis and positive ERα status. When these epigenetic processes are targeted, oncogenic signals such as Wnt signaling are normally regulated (Figure 1).

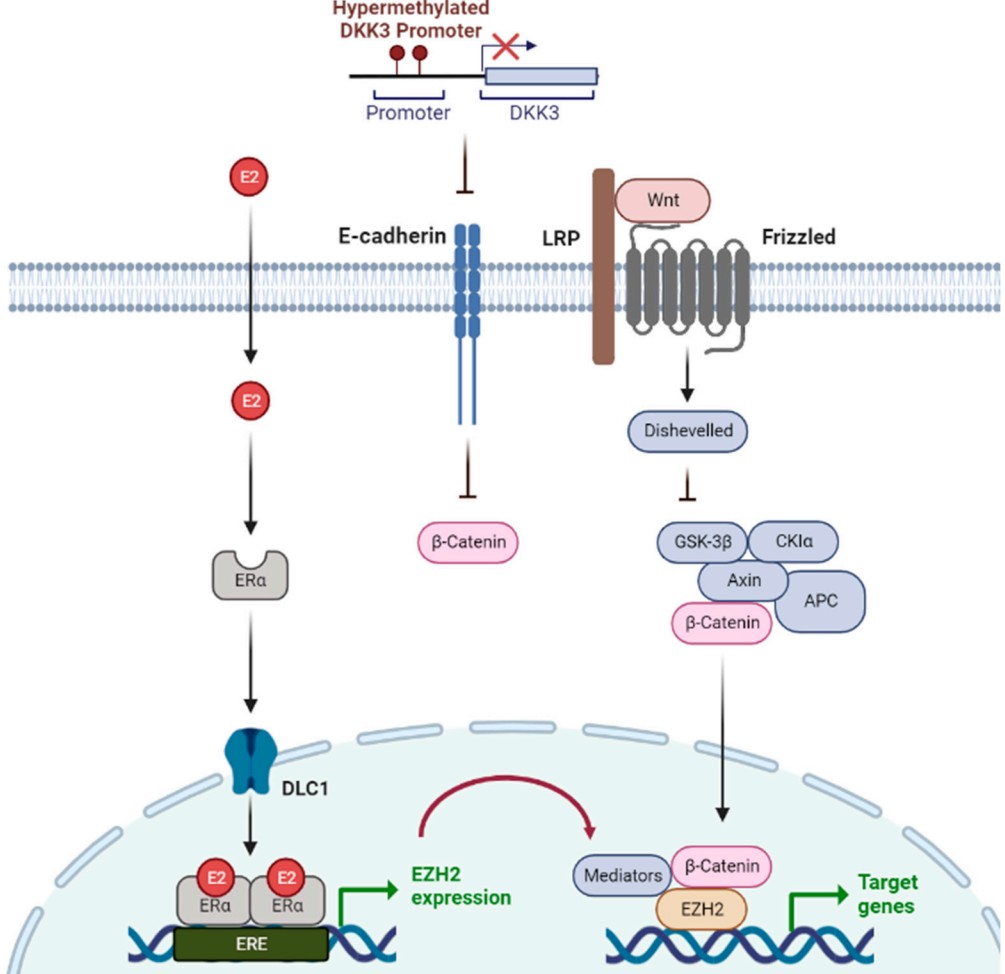

**Figure 1.** Epigenetic mechanisms regulated via 17-beta estradiol (E2) and Wnt signaling in breast cancer. E2 is a type of the hormone estrogen, which can be a risk factor for breast cancer. Wnt signaling is one of the critical signaling pathways in tumor progression. EZH2 is expressed by E2-estrogen receptor (ER) binding and connects estrogen signaling and Wnt signaling. Wnt signaling facilitates oncogenesis when Wnt antagonist gene, DKK3, is silenced via promoter hypermethylation.

### 3.3. Epigenetic Modulation during Epithelial to Mesenchymal Transition (EMT) in Breast Cancer

The EMT plays a critical role in cancer metastasis. SPCA2, a calcium-ATPase, is expressed at an abnormally low levels in TNBC and is associated with the EMT of tumor cells. HDACi promotes mesenchymal to epithelial transition in TNBC cell lines by enhancing SPCA2 expression, suggesting that abnormal epigenetic regulation of SPCA2 is related to breast cancer metastasis and poor prognosis. Another epigenetic regulatory mechanism of metastasis in breast cancer is *CDH1* promoter methylation [29]. *CDH1*, a tumor suppressor gene that encodes E-cadherin, mediates cell-to-cell adhesion between neighboring cells. Low E-cadherin expression mediates the EMT process during tumorigenesis, leading to breast cancer invasion and metastasis. In invasive breast cancer, *CDH1* is abnormally highly methylated, and E-cadherin expression is markedly downregulated, rendering it more invasive. Hence, 40.9% of the samples derived from patients with primary breast cancer demonstrate *CDH1* methylation, resulting in metastasis and poor prognosis [30].

*CDH1* has also been implicated in stem cell-like properties of breast cancer. Breast cancer stem cells play a critical role in tumorigenesis and recurrence; therefore, it is important to aim for complete eradication of breast cancer stem cells. Hypermethylation of *CDH1* promotes stem cell-like properties of tumor cells. Hence, some modulators such as phenethyl isothiocyanate (PEITC) can destroy tumor cells by eliminating CSC-like properties by demethylating *CDH1* [31].

## 4. Epidrug

Epigenetic modifications alter chromatin structure and gene expression due to both external (i.e., stress, nutrition) and internal factors (i.e., genes and aging). Mechanisms that mediate epigenetic changes include histone modifications, DNA methylation, deregulation of non-coding RNA, and interactions with proteins or nucleic acids. These epigenetic dysregulation mechanisms can affect multiple pathways, including EMT, Hippo signaling, the p53 pathway, AMPK signaling, and cellular senescence, eventually leading to mutations and cancer. Epidrugs are small-molecule inhibitors that decrease enzymatic activity and target enzymes that contribute to the regulation of aberrant epigenetic alterations [32,33].

### 4.1. DNMT Inhibitors (DNMTis)

DNMTis, such as azacitidine and decitabine, are the most optimal therapeutic combination regimens and are still widely used. It is an enzyme-targeted epidrug of DNMTs that intercalates between DNA base pairs and inhibits the methylation of CpG dinucleotides. DNMTis can be classified into three types: nucleoside analogs, non-nucleoside compounds, and natural compounds. DNMTis generally induce hypomethylation through cell division, leading to increased tumor suppressor expression. In addition, non-nucleoside compounds can inhibit DNMT activity. For example, the demethylating agent hydralazine serves as a transcriptional reactant for tumor suppressor genes. Additionally, natural compounds can block the activity of DNMTs. Isoflavones, which are natural compounds, directly block DNMT and reactivate genes silenced by methylation [12,34].

In mammals, DNMTs are classified as DNMT1, DNMT2, and DNMT3. DNMTi-mediated inhibition of DNMT1 has been well-studied as an epigenetic treatment strategy for cancer. Azacitidine and decitabine have been approved by the Food and Drug Administration (FDA) and are used to treat certain cancers [35]. For instance, decitabine, a deoxycytidine analog, is an irreversible DNMT1 inhibitor that promotes apoptosis by influencing the cell cycle. It can also be used as an FDA-approved drug to treat the myelodysplastic syndrome. Azacitidine, a pyrimidine nucleoside analog, induces DNA demethylation by inhibiting the activity of DNMT1. It has been approved by the Food and FDA for the treatment of the pre-leukemic myelodysplastic syndrome. Briefly, azacitidine decreases abnormal DNA methylation and exhibits antitumor activity that prevents cancer development [36]. However, first-generation DNMTis are harmful to the human body at high concentrations. Hence, patients have to be exposed to such compounds for a long time at low concentrations. Therefore, the development of second-generation DNMTis is required to overcome poor sensitivity. To date, the development of DNMTi for cancer treatment has been actively studied by establishing DNMT suppression and gene reactivation strategies [37].

### 4.2. HDAC Inhibitors

The balance between histone acetylation and deacetylation by HATs and HDACs is normally well-regulated, but is often dysregulated in cancers [38]. HDAC inhibitors, which target HDAC enzymes and promote histone acetylation, are emerging as novel anti-cancer agents. HDACs act as proteolytic enzymes that deacetylate acetylated histones and inhibit gene transcription. When HDACs are overexpressed in cancer cells, they tighten loose nucleosomes and suppress the expression of tumor suppressor genes, which helps cancer cells multiply, metastasize, and survive. HDACis suppress HDACs to activate gene transcription, eventually leading to cell death. However, HDACis exhibit many associated

side effects, including hematological toxicity, cardiac arrhythmia, and weight loss. Because of these side effects, HDACis with low toxicity and high efficiency should be selected for various cancer types. Representative HDACis currently approved by the FDA are vorinostat, abexinostat, and panobinostat [39,40].

Vorinostat inhibits the activities of HDAC1, HDAC2, HDAC3, and HDAC6 by reversibly binding to $Zn^{2+}$ at the active site of the enzyme [41]. It exhibits antitumor activity via various targeted regulatory mechanisms. In breast cancer, vorinostat and PKF118-310, a Wnt-β-chain protein blocker, induces the differentiation of cancer stem cells and suppresses EMT to reduce the number of breast cancer stem cells [42]. Abexinostat inhibits the activity of HDAC1, HDAC2, HDAC3, HDAC6, HDAC8, and HDAC10 to treat non-cell cycle-specific cytotoxicity and 4-line follicular lymphoma. It shows higher safety and clinical efficacy than existing HDACi. Abexinostat reduces the expression and repair ability of *RAD15*, a gene involved in homologous recombination for anticancer treatment. It also inhibits EMT by aberrantly regulating calcium refraction, which induces apoptosis in breast cancer [43,44]. Panobinostat, which has been approved by the FDA to treat multiple myeloma in combination with other drugs, can induce apoptosis by affecting only cancer cells and not normal cells [45]. Currently, the development of many HDACi epidrugs is still in progress, but they have limitations, such as off-target effects, low selectivity, and low efficacy (Figure 2).

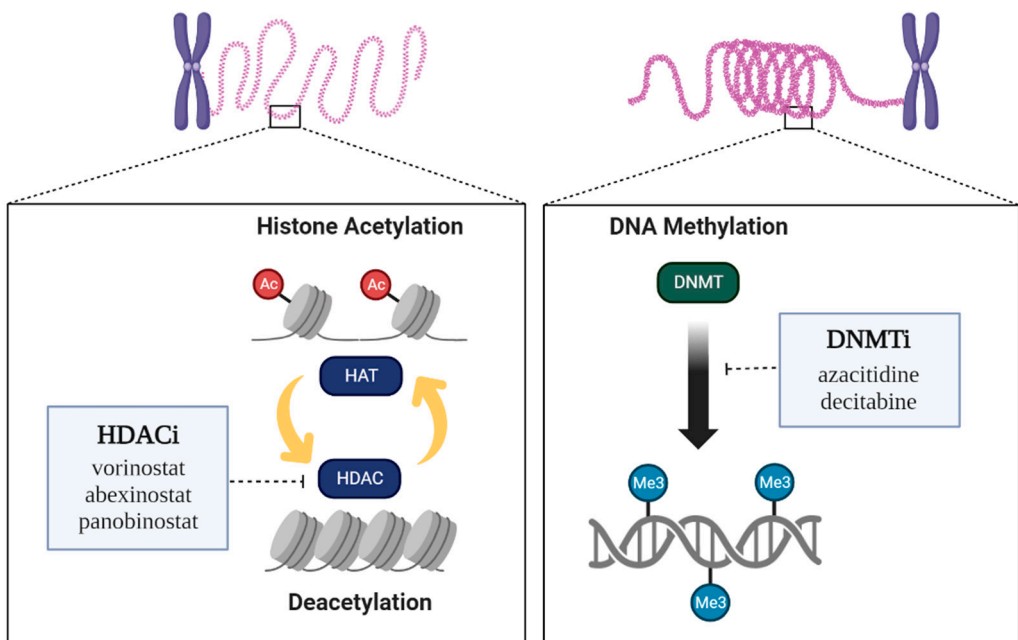

**Figure 2.** Epigenetic regulation by enzymes and Epidrugs. Two epigenetic targets or mechanisms of epidrugs are known. HDAC inhibitors act at histone deacetylases (HDACs), which inhibit histone acetyltransferases (HATs). DNMT inhibitors act at DNA methyltransferases (DNMTs).

*4.3. Recently Developed Epidrugs*

Epidrugs, which target and stimulate immune responses, have garnered interest as a promising strategy for increasing the effectiveness of existing treatments and overcome resistance to immuno-anticancer drugs. Modulation of the inflammatory response via epigenetic machinery can expand the clinical use of epidrugs to restore and enhance the responses of patients to the immunotherapy [46]. Intriguingly, trichostatin A increases tumor cell immunogenicity, which induces an antitumor effect. Hydroxamic acid, LBH589, inhibits melanoma cells by inhibiting tumor growth and enhancing the immunogenicity [47]. Moreover, tazemetostat which has been ongoing phase II is a EZH2 inhibitor and works on different kinds of lymphomas and solid tumors [48,49]. Targeted protein degradation through Proteolysis Targeting Chimeras (PROTACs) has also been utilized to regulate

epigenetic status [50]. PROTAC has two ligands that bind to both proteins of interest and E3 ubiquitin ligase recruiting element [51]. Recently, Schiedel et al. demonstrated that PRROTACs can target epigenetic eraser proteins including sirtuins [50]. Several clinical trials have suggested the synergistic effects of epidrugs and various combination therapies, such as immunotherapy, radiation therapy, and endocrine therapy [52]; however, epidrugs are still limited to hematological malignancies with low efficacy.

## 5. Epidrugs in Breast Cancer

Mutations in histone methyltransferases stimulate the generation of epidrugs that target a wide range of chromatin regulators [53]. Epigenetic agents can induce cell death in response to endocrine therapy. Tamoxifen treatment induces autophagy, which increases cancer cell death. However, it can also cause tamoxifen-resistant breast cancers. HDACi treatment can induce the expression of pro-apoptotic proteins, including BAX and BAK. Therefore, it can be used with the combination of HDACi and tamoxifen, which mainly redirects ER-positive breast cancer cells to apoptosis. This strategy could be used to initiate a better clinical trial with a combination of HDACi, exemestane, and tamoxifen.

Iadademstat (ORY-100) is a lysine-specific demethylase 1 (LSD1) inhibitor that is currently being examined in a phase II trial to treat AML [54]. Luminal-B breast tumors are ER+ and HER2+ or negative subtypes. Targeting SOX-2 driven cancer stem cells has demonstrated the clinical promise of iadademstat as an anti-ti-SOX2 epigenetic breast cancer treatment, especially in the SOX2-rich luminal-B HER2+ type. LSD1 is a selective epigenetic target in SOX2-expressing cancer treatment, particularly in SOX2-rich carcinomas associated with small cell lung cancer (SCLC), ovarian cancer, and cervical cancer [55]. Iadademstat treatment of patients with luminal-B breast cancer and mammospheres significantly reduced SOX2 expression, suggesting the selective targeting of SOX2-driven CSCs. LSD1-targeted epigenetic methods, including iadademstat for the prevention of breast and ovarian cancers, have great potential for cancer treatment [56].

### 5.1. DNMTi in Breast Cancer

DNMTis have strong potential for cancer therapy. Recently, the use of DNMTis has been emphasized in the field of immune oncology. DNMTi improves tumor immunogenicity by secreting cytokines via the stimulation of immune cells. DNMTi is a promising therapeutic agent to treat cancer and exerts anti-tumorigenic effects against breast cancer [57]. Yanrong Su et al. investigated the possibility of using DNMTi to target EMT to treat TNBC. It was shown that highly aggressive TNBC cells were reprogrammed to less aggressive TNBC cells when subjected to EMT. DNMTi also exhibited antitumor effects by inhibiting cell proliferation, such as inducing cell cycle arrest. These results support that DNMTi can be reported as a promising therapeutic agent showing antitumor effect [58]. Decitabine (5-aza-2′-deoxycytidine), a demethylating agent, is an FDA-approved DNMTi. Decitabine fusions to DNA irreversibly bind to DNMTs and take up enzymes on DNA, inducing DNMT failure. Decitabine has been proven to be effective in the treatment of hematological malignancies. In breast cancer, decitabine-treated TNBC showed high sensitivity. This led not only to proteasomal-dependent DNMT1 degradation but also to DNMT3A and DNMT3B degradation at low concentrations. Moreover, patient-derived xenograft (PDX) organoids have shown significant effects on tumor growth inhibition with low concentrations of decitabine [59]. Further studies demonstrated that decitabine induced autophagy in breast cancer cells, as indicated by an increase in the autophagy marker LCB-II. It is also effective in promoting autophagy in ovarian cancer [60].

Azacitidine also has the potential as a novel therapeutic agent for preclinical breast cancer treatment. Azacitidine treatment inhibits breast cancer brain metastasis. Under both in vitro and in vivo conditions, azacitidine suppresses the Wnt signaling pathway, cell migration, cell invasion, and tumorigenesis in brain-colonizing cells [61]. In another study, combination treatment with azacitidine and vorinostat resulted in upregulation of PD-L1 mRNA expression [62].

Guadecitabine has been suggested as an alternative to conventional DNMTis, including decitabine and azacytidine because it can be translated into front-line therapy. Guadecitabine increases PD-L1 expression and MHC class I expression and suppresses tumor cell proliferation. Moreover, early treatment with guadecitabine inhibits tumor growth initiation in vivo [63]. In addition, guadecitabine in combination with HDACi can reprogram aggressive TNBC cells. Guadecitabine also upregulates MHC class I and II in TNBC and promotes the recruitment of CD8 + T cells to the tumor microenvironment in vivo [57]. Therefore, guadecitabine might be a good anti-tumor agent for patients with breast cancer.

Liraglutide is a well-known diabetes drug, but it can act as a DNMTi in breast cancer in vitro and in vivo in Ehnlich mouse tumor models. The results revealed that liraglutide reduced cell viability, migration, and DNMT activity [64]. It is effective in promoting anti-proliferative activity and deteriorating migration and motility of mesenchymal breast cancer cell lines. Importantly, unlike previous drugs, hypermethylation of the *DNMT* gene after treatment with liraglutide was not observed. The combination of half-dose chemotherapeutic agents and liraglutide significantly reduces side effects, such as toxicity and reproductive dysfunction [65]. Therefore, liraglutide could be evaluated as a new adjuvant to improve breast cancer treatment.

## 5.2. HDACi in Breast Cancer

HDACs are essential for the key genes involved in cancer cell growth and survival. Therefore, targeting HDACs using HDACis contributes to an efficient therapeutic strategy for cancer cells. HDACi can be classified as hydroxamic acids (hydroxamates), short-chain fatty acids, benzamides, cyclic tetrapeptides, and sirtuin inhibitors [66]. In cancer clinical trials, combination therapy with HDACis is commonly used.

Vorinostat was the first HDACi approved by the FDA [67]. Vorinostat inhibits the proliferation of TNBC cells by upregulating miRNA expression, which induces the expression of tumor suppressor genes. Moreover, the combination of vorinostat with simvastatin (a cholesterol-lowering drug) can induce apoptosis via the interruption of Rab7 prenylation and suppression of autophagosome-lysosome fusion in TNBC [68]. Combination treatment also showed apoptotic effects by inhibiting Rab7 prenylation in xenograft mice in vivo. This indicates that Rab7 is a promising drug target for the combination of vorinostat and simvastatin. In another study, the combination of vorinostat and letrozole (an aromatase inhibitor) led to inhibited breast cancer cell proliferation, apoptosis, and differentiation of peripheral blood mononuclear cells into osteoclasts. This combination may minimize the risk of osteoporosis in patients with breast cancer. Additionally, treatment with vorinostat along with immune checkpoint blockers, such as PD-1 and CTLA-4, can stimulate tumor apoptosis and regression in TNBC [69].

Transwell invasion, migration, and wound healing assays have revealed the invasion and migration abilities of breast cancer cells are significantly inhibited by trichostatin A (TSA) treatment [70]. TSA, a potent pandeacetylase inhibitor, can modulate the transcriptional action of ERβ in ERα-negative BC, inducing an HR-negative breast cancer cell response to tamoxifen. This renders TNBC cells more sensitive to tamoxifen activity [71]. Panobinostat, an FDA-approved HDACi, can improve histone acetylation of H3 and H4, the cell cycle, and apoptosis in breast cancer. In an in vivo study, it inhibited proliferation and enhanced histone acetylation in TNBC cells [72]. Panobinostat can also re-express silenced ERα in TNBC and induce sensitivity to tamoxifen. Thus, the combination of panobinostat and trastuzumab has been indicated as an anti-HER2 treatment [73]. Another combination treatment with panobinostat and letrozole inhibited aromatase expression in hormone-responsive breast cancer cells [74], suggesting that these combination therapies are likely to target hormone receptor-positive/aromatase-positive breast cancer.

Varprobic acid (VPA) has recently emerged as a promising cancer treatment. It has been used successfully for over 50 years to treat epilepsy, bipolar disorder, and schizophrenia [75]. VPA exhibits potent antitumor effects in vitro and in vivo, either alone or in

combination with demethylating cytotoxic agents, facilitating favorable clinical trial outcomes. VPA inhibits the growth of HER2+ breast cancer cells by upregulating the p21 WAF1 expression [75]. It also induces apoptosis and histone H3 acetylation by interrupting the activity of hsp90 [76]. The combination of VPA and 5-aza-2'-deoxycytidine (a DNA methyltransferase inhibitor) has the ability to reactivate transcription of the RA receptor β2 tumor suppressor gene in breast cancer [77], leading to induction of apoptosis in breast cancer stem cells.

The efficacy of HDACi as a single agent in solid tumors is not always favorable. Therefore, co-treatment with other therapies, such as HDACi, chemotherapy, hormone therapy, and immunomodulatory agents is recommended. HDACi can reestablish abnormal acetylation of proteins associated with cancer pathways and reactive tumor suppressor genes, leading to cell cycle arrest and apoptosis of cancer cells [78]. HDACi resistance is also a critical impediment to HDACi therapy. To improve the clinical efficacy of HDACi, combination therapies have shown significantly more potent effects compared to monotherapy. Combination therapy can overcome HDACi resistance in an ideal manner. Identification of new selective HDACis and predictive biomarkers for HDACi therapy and elucidation of detailed mechanisms of HDACi will aid in supporting the clinical use of HDACi in the breast cancer treatment.

### 5.3. Combination Therapy with Epidrugs in Breast Cancer

Previous studies have investigated the efficacy of DNMTi and HDACi in breast cancer. However, they have shown limited efficacy at the maximum tolerated dose. Therefore, epidrugs have been used in combination with cytotoxic agents, radiation therapy, targeted therapy, and hormonal therapy for breast cancer [79]. However, the results of clinical trials have been unsatisfactory due to systemic toxicity and limited efficacy. Therefore, it is necessary to discover appropriate epigenetic biomarkers for personalized approaches and the targeted delivery of epidrugs. In particular, HDACi treatment for ER+ cells showed increased antiproliferative endocrine therapy activity. The azacytidine–entinostat combination and HDACi therapy alone showed ER re-expression and effective resistance to anti-estrogen therapy in ER-breast cancer [80]. Moreover, Bromodomain and Extra Terminal motif (BET) inhibitor, JQ1, alone or in combination with selective ER downregulation-promoting molecules, suppressed the growth of tamoxifen-resistant cells [81]. In addition, a synergistic effect of combination therapy has been demonstrated in TNBC. For instance, the synergistic efficacy of HDACi and anti-HER2 therapy using trastuzumab has been investigated in a clinical study [73]. Thus, an appropriate combination of drugs can target oncogenic mechanisms.

### 5.4. Epidrugs with Nanotechnology

The instability, toxicity, and off-target effects of epidrugs are major factors that prevent their success in solid tumors. Nanotechnology can provide a method to target therapies directly and selectively in cancer cells. This technology enables safer and more effective delivery of epidrugs. Furthermore, cancer nanotechnology can reduce systemic toxicity by improving pharmacokinetics and selectively targeting and delivering anticancer drugs to tumors. For instance, nanoparticles, including membrane-camouflaged, exosome-disguised, albumin, and lactoferrin nanoparticles, are novel nanotechnologies for targeting tumor cells and regulating the tumor microenvironment in breast cancer. Epigenetics-based nano-delivery enhances apoptosis and interferes with proliferation and migration [82]. Nanomedicine can be used to overcome the low efficacy of treatment-resistant tumor therapy by combining it with new-generation epidrugs [16]. This suggests that applications of nanotechnology in medicine offer new opportunities to enhance epidemic delivery, improve stability and solubility, and minimize off-target effects.

## 6. Trends in Clinical Trials of Epidrugs in Breast Cancer

*6.1. Clinical Trials in Breast Cancer*

Preclinical studies have demonstrated that epidrug DNMTi can reduce the tumorigenicity of cancer stem cells by suppressing stemness and differentiation-related gene expression. HDACis inhibit the activity of cancer stem cells by targeting several essential genes involved in cancer stem cell maintenance, such as those encoding β, γ-catenin, Stat3, and Notch1, thereby reducing tumor development [83]. Clinical reactions are typically non-cytotoxic and were observed in patients receiving low concentrations of DNMTi- and PD-1-conjugated therapeutic drugs. These inhibitors can restore the TME to a normal state in patients with colorectal cancer. However, HDACis in clinical trials are only effective in hematological malignancies, and clinical trials for solid tumors have shown no significant effects [84]. A phase II clinical trial for the treatment of breast cancer with CC-486, hypomethylating agent (HMA), and durvalumab has been conducted. Phase II clinical trials have shown only marginal clinical responses. Anti-PD-1, anti-CTLA4, or a combination of the two, have shown tumor reduction and increased survival in HER2/neurogenetic breast cancer models in clinical trials. VPA is an antiepileptic drug that is normally used for the treatment of progressive prostate and breast cancers. It selectively inhibits class I HDACs and reduces tumor growth and metastasis in vivo. In another study, using a mouse model of malignant pleural mesothelioma (MPM), multiple drugs, including decitabine 2 and VPA, showed an anti-tumor immune response by mediating the expression of cancer-testis antigen (CTA) [85]. In clinical trials, VPA has been combined with decitabine to study the immunogenic ability of the new HDACi. Additionally, mRNA expression of PD-L1, CTA, and retinoic acid-inducible protein I (RIG-I) was observed in MPM cells [48].

Hydralazine can induce demethylation and reactivate tumor suppressor genes as a treatment for hypertension. This can enhance the effectiveness of biological and chemical treatments. In phase I, a clinical trial was conducted to confirm the safety of the dose and standard cytotoxic chemotherapy in patients with breast cancer. The results of phase I indicated that hydralazine was well tolerated and had no side effects in chemotherapy at doses ≤ 200 mg. Zambrano et al. showed the demethylation of up to 52% of the promoter region in selected tumor suppressor genes at a safe dose range to avoid toxicity. Additional studies on the combination of hydralazine and standard cytotoxic chemotherapy are required, as they seem promising strategies to improve the efficacy in phase II [86,87].

*6.2. Limitations and Prospects*

Some challenges are associated with the main strategies of epidrugs with respect to cytotoxicity, tolerance, selectivity, and potency. When treated with epidrugs alone, one-third of patients with early-stage ER+ breast cancer develop resistance to treatment and drug resistance in TNBC [88,89]. Three strategies can be used to overcome the low efficacy of single epidrug targets: multiple-medication therapy (MMT), multi-compound medication (MCM), and multi-target-directed ligand (MTDLs) approaches. Such a strategy is expected to have high efficacy, including superior therapeutic effects, reduced side effects, and reduced potential for drug resistance [90]. MMT increases chromatin accessibility to DNA-damaging chemotherapeutic agents and enhances the efficacy of drug responses through multiple drugs. For instance, CpG island methylation induces drug resistance; however, in the MMT strategy, the combination of zebulin and decitabine helps demethylation [91]. MMT can increase the therapeutic effect at a low dose by minimizing drug resistance and side effects compared with single drugs, allowing a variety of drug doses to be selected for personalized treatment. Similarly, MCM involves combination of two or more active principles, each with a unique goal, and the strategy is utilized as "polyvalent pills" [92]. Finally, the MTDLs strategy signifies the development of a single molecule that can simultaneously interact with different targets. MTDLs are hybrid molecules that use the mechanism of HDACis in conjunction with that of other drugs to target and act against cancer [93].

An alternative method that can compensate for the disadvantages of a single drug is the multi-drug method, which is a more effective strategy. The triple combination of HDACi, romidexin, cisplatin, and nivolumab showed high efficacy in refractory metastatic TNBC. However, multidrug therapies do not always function positively. For example, there was no significant difference in the overall response rate and progression-free survival with atezolizumab alone, and adverse effects were observed with combination therapy [83]. Therefore, new approaches to treat epidrugs in combination with various therapies to enhance the antitumor effects must be developed.

## 7. Conclusions

As epigenetic regulatory mechanisms are closely related to cancer progression, epigenetic reprogramming of cancer cells can be a potentially powerful therapeutic approach. Epigenetic drugs, referred to in this review as 'epidrugs', usually act on essential enzymes for processes that regulate epigenetic modifications, such as DNMTs, HDACs or BETs (Table 1). Owing to the importance of epigenetic alterations in cancer and the possibility of using chemotherapy and epigenetic drugs in combination, the FDA has approved several epigenetic therapies for the treatment of cancer. Breast and ovarian cancers, which are representative gynecological cancers affecting women, are one of the main targets of various epidrugs. However, some challenges are still associated with the use of single-agent epidrugs; therefore, understanding the benefits and risks of drug combinations can help us leverage epidrugs and provide a different perspective on cancer treatment. In addition, attempting to find more epigenetic targets that can be utilized as epidrugs in cancer will also be helpful for epidrug development. Recently, next-generation sequencing data have provided information on genetic mutations and epigenetic modifications. Research on epigenetic modifications in cancer will allow us to design rational interventions for anti-cancer strategies. Multidrug therapies in combination with other compounds are promising cancer treatments for tumor remission, reduced resistance to chemotherapy, and reduced side effects. We must focus on finding novel epigenetic targets and an optimized treatment combination with a high synergistic effect and absence of side effects to treat breast cancer.

**Table 1.** Representative epidrugs for cancer treatment.

| Subtype (Class) | Epidrug Name | Effect | Combination | FDA-Approval | Reference |
|---|---|---|---|---|---|
| DNA methyltransferase inhibitor (DNMTi) | 5-azacitidine | Treat myelodysplastic syndrome | Vaprobic acid Entinostat | Yes | [36] |
| | Decitabine | Treat myelodysplastic syndrome and acute myeloid leukemia (AML) | Zebulin | Yes | [36] |
| | Guadecitabine | Treats myeloid malignancies and inhibits tumor growth | Pembrolizumab | Yes | [93] |
| | Isoflavones | Treats prostate cancer and inhibits thyroid hormone production. | Resveratrol | No | [12,33] |
| | Hydralazine | Treats high blood pressure and heart failure | Isosorbide dinitrate | Yes | [12,33] |

**Table 1.** *Cont.*

| Subtype (Class) | Epidrug Name | Effect | Combination | FDA-Approval | Reference |
|---|---|---|---|---|---|
| Histone deacetylase inhibitor (HDACi) | Vorinostat | Inhibits the proliferation of TNBC cell; Treat cutaneous T-cell lymphoma | PKF118-310 Simvastatin letrozole | Yes | [41,68] |
| | Abexinostat | Treats non-cell cycle-specific cytotoxicity and 4-line follicular lymphoma. | Pembrolizumab | Yes | [43] |
| | Panobinostat | Treats multiple myeloma in combination with other drugs. | Trastuzumab Letrozole | Yes | [45,72] |
| | Trichostatin A | Inhibits invasive and migratory abilities of breast cancer cells | Cisplatin Gemcitabine Doxorubicin | No | [70] |
| Lysine-specific demethylase 1 inhibitor (LSD1i) | Iadademstat | Acts as an immunomodulator and candidate for therapeutic combinations in leukemia or some solid tumors. | Azacitidine Venetoclax | Yes | [56] |

**Author Contributions:** Conceptualization, A.K., K.M., H.K., W.K. and H.Y.; writing—original draft preparation, A.K., K.M., H.K., M.P., S.C., W.K. and H.Y.; writing—review and editing, A.K., K.M., H.K., M.P., S.C., W.K. and H.Y.; supervision, W.K. and H.Y.; funding acquisition, H.Y. All authors have read and agreed to the published version of the manuscript.

**Funding:** This research was funded by Brain Korea 21 (BK21; # M2022B002600003).

**Conflicts of Interest:** The authors declare no conflict of interest.

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
