# Peer review of "Epigenetic Regulation in Breast Cancer: Insights on Epidrugs"

_2075-4655, 2023_

Round 1

Reviewer 1 Report

This is a well-written manuscript titled “Epigenetic Regulation in Breast Cancer: Insights on Epidrugs” that discuss the importance of epigenetic regulation and the potential of epigenetic-targeted therapy using epidrugs in breast cancer. The manuscript is well-written by the author and covers very detailed previous and recent reports on epigenetic regulation in Breast cancer.   The manuscript will be informative for readers to understand epigenetic regulatory mechanisms in breast cancer progression and epigenetic-targeted therapy. The manuscript can be accepted in its present form.

Author Response

We appreciate the reviewer’s summary and valuable comments on our manuscript.

Reviewer 2 Report

This is a nice review by the authors. However in order to make it more comprehensive, the authors should address the following points:

1. The authors need to include separate sections for the histone methyl transferases, and histone acetyl transferases. 

2. The different types and patters of drugs used currently: may be add a paragraph about PROTACS. 

3. The authors have ignored the HMT and HATs, please elaborate these in the review.

4. please add 1-2 figures which would talk about the epidrugs, and describe the article in a better way. 

5. Figure 2 is misleading, although the authors claim these changes in modifications on histones and DNA, but the figure does not differentiate the histone and DNA modifications.

6. Would activators of HATs also called as epidrugs, if so, explain and give examples in the text, and similarly, histone. Please reframe your abstract accordingly.

Thank you for your great efforts.

Reviewer 3 Report

Epigenetic Regulation in breast cancer: Insights on epidrugs

Kim A., et. al.

Summary

Kim et al try to summarize the state of affairs in epigenetic therapy, in general. They take stock of epigenetic regulatory mechanisms and how some of these could be tackled in the context of breast cancer.

Comments:

1.      The introduction is haphazardly written. There is no continuity or flow of various concepts discussed.

2.     The title “2.1 Noncoding RNA-mediated epigenetic regulation” is a misleading. In the section (lines 98-118) the authors mention miRNAs and lncRNAs controlling growth factor driven signaling pathways. While there can be a place and time where non-coding RNAs are termed ‘epigenetic’, these RNAs are by no means epigenetic modifiers or regulators.

3.     Lines 128-130: All the transcriptional co-regulators listed here can in fact bind to DNA even in the absence of E2. The transcriptional output of many of these co-regulators is enhanced by ER-alpha. – Neither of these is evident from how it is written in these lines.

4.    Line 131: Needs Reference.

5.     Line 131-133: Please summarize study in reference 25 more clearly and succinctly.

6.     Line 138: DKK3 methylation: please state that it is DKK3 promoter methylation.

7.     Lines 150-167: “2.3 Epigenetic modulation in epithelial to mesenchymal transition (EMT) of breast cancer”: there are no epigenetic mechanisms listed. The roles SPCA2 and CDH1 are listed here. While the statements made in this section are true, they are causative. What epigenetic factors regulate SPAC2 expression in TNBC? CDH1 promoter hypermethylation is a prognostic factor of TNBC, and to some extent reversible by PITC treatments.

8.     Section 3: Epidrug

Any drug – targeting epigenetic mechanisms or not – can cause systemic effects on

the cells or animals or individuals.

a.     In each of the sections pertaining to specific class of epigenetic modifications, the authors could list what kinds/classes of drugs worked or not.

b.     In section 3.3: the authors could attempt to summarize various clinical trials and/or novel molecules published by various groups and/or pharmaceutical companies.

c.     There are other histone modifications such as histone arginine methylation, histone lysine methylation, histone lysine demethylation, etc. The authors should summarize what is known about these. Or at least state clearly if/why the focus of the current review excludes these other modifications.

9.     Line 267: Please refer to the actual study. “the current study” is incorrect.

Round 2

Reviewer 2 Report

Please add some more recent citations.

add more sentences for Prozac's

Reviewer 3 Report

The authors have made good improvements to the manuscript.

I recommend publishing it.

Thank you.